electrical engineering

wireless communications, orbital angular momentum multiplexing, spiral phase plates

**Author for correspondence:**
B. Allen
e-mail: ben.allen@networkrail.co.uk

# Experimental evaluation of 3D printed spiral phase plates for enabling an orbital angular momentum multiplexed radio system

B. Allen[1,2], T. Pelham[3], Y. Wu[4], T. Drysdale[5], D. Isakov[6], C. Gamlath[3], C. J. Stevens[1], G. Hilton[3], M. A. Beach[3] and P. S. Grant[4]

[1]Department of Engineering Science, University of Oxford, Parks Road, Oxford OX1 3PJ, UK
[2]Network Rail, The Quadrant:MK, Elder Gate, Milton Keynes MK9 1ER, UK
[3]Department of Electrical and Electronic Engineering, University of Bristol, Woodland Road, Bristol BS8 1UB, UK
[4]Department of Materials, University of Oxford, Parks Road, Oxford OX1 3PH, UK
[5]Institute for Digital Communications, University of Edinburgh, Edinburgh EH9 3FG, UK
[6]WMG, International Manufacturing Centre, University of Warwick, Warwick CV4 7AL, UK

BA, 0000-0002-6308-8383

This paper evaluates the performance of three-dimensionally (3D) printed spiral phase plates (SPPs) for enabling an orbital angular momentum (OAM) multiplexed radio system. The design and realization of the SPPs by means of additive manufacturing exploiting a high-permittivity material is described. Modes 1 and 2 SPPs are then evaluated at 15 GHz in terms of 3D complex radiation pattern, mode purity and beam collimation by means of a 3D printed dielectric lens. The results with the lens yield a crosstalk of −8 dB for between modes 1 and −1, and −11.4 dB for between modes 2 and −2. We suggest a mode multiplexer architecture that is expected to further reduce the crosstalk for each mode. An additional loss of 4.2 dB is incurred with the SPPs inserted into the communication link, which is undesirable for obtaining reliable LTE-based communications. Thus, we suggest: using lower loss materials, seeking ways to reduce material interface reflections or alternative ways of OAM multiplexing to realize a viable OAM multiplexed radio system.

# 1. Introduction

The demand for radio spectrum continues unabated. This is driven by more users wanting ubiquitous access to data-hungry services such as recorded or streaming media and collaborative games. Telecommunications systems support these services by transporting data between networks of data centres, and to/from users. Free space communication systems can be deployed at a lower cost than cabled solutions but support lower data rates. However, in 2012, Tamburini *et al.* [1] proposed the possibility of using orbital angular momentum (OAM) radio to multiplex several data streams on the same frequency, polarization and time. The technique opened the potential to significantly increase the data rates of point-to-point wireless links and to challenge that achievable by some optical fibre connections. Significantly, in 2018, a 1 Tb s$^{-1}$ data rate wireless link was demonstrated wirelessly at 26 GHz by using OAM modes to multiplex 10 channels [2]. More recently, 2 Tb s$^{-1}$ has been demonstrated using 20 channels [3]. Despite these achievements, OAM radio continues to attract much research attention as several unanswered questions remain about maximum practical performance limits in the radio domain. Improvements in aspects such as link distance, crosstalk and optimum network architecture are sought by the research community [4].

When OAM modes are implemented using circular antenna arrays, OAM radio may be considered as a type of multiple input multiple output (MIMO) communications systems [5]. However, OAM radio links differ from conventional MIMO because it is only well suited to line-of-sight links and does not rely on path diversity for attaining a performance gain. As is the case for MIMO, OAM radio links have the potential to support multiple parallel data streams by sending each stream on a separate 'mode'. For OAM, however, a mode relates to the radiated spatial amplitude and phase patterns as defined by the family of the Laguerre–Gaussian (LG) beams [6], where the phase signatures are mutually orthogonal [7]. Due to the phase trajectory of each mode having a 'twist', they have a topological charge and hence are referred to as OAM modes [8]. Ideal amplitude and phase patterns of modes $l = 2$ and $l = -1$ are shown in figure 1. Mode 0 (not shown) is the conventional Gaussian mode used in traditional point-to-point communications links, i.e. the radiated beam profile is Gaussian. Modes with $|l| > 0$ will always have a zero amplitude in the centre resulting in a ring or 'doughnut' shaped amplitude pattern in their far field, while their phase has a 'vortex' pattern. For these modes, the amplitude profiles differ in terms of the radius of the 'doughnut'. The effect of the vortex is that the beam is heavily divergent, e.g. a divergence of 14° would result in a 25 m radius at 100 m. This property severely limits the practical link distance. This is mitigated by means of placing a collimating lens such as the one described in [9] in the path of the beam. The phase profile for mode $l = 2$ in figure 1 shows a linear increment of phase around the central discontinuity in a clockwise direction, where the phase has a total of $4\pi$ radians around a circle. For the case of mode $l = -1$, the phase increments in a counter-clockwise direction and increments $2\pi$ radians around a circle. OAM modes differ from polarization, in that polarization may be likened to spin angular momentum (SAM), which is usually an independent effect to OAM [10]; hence, the two phenomena may both be exploited for enhancing communications link performance.

Wireless communications employing OAM multiplexing may exploit available OAM modes as a means of providing an additional degree of freedom when allocating resources. There are two benefits to adopting additional degrees of freedom in this way: first, increased total transmission power without the inefficiency of having to combine the power of multiple millimetre-wave sources in a single-mode link; and second, increased spectral efficiency for the same total transmission power compared to a single-mode link. Ideally, each OAM mode is independent of the others, but in practice a level of inter-modal crosstalk (the difference in signal level between any pair of modes) occurs between each mode because of: coupling of RF signal paths; generator imperfections resulting in transmission of non-ideal modes; detector imperfections resulting in reduced discrimination between received modes; and multipath interference caused by poor link positioning or nearby objects causing field perturbations.

In addition to crosstalk, the divergence of the modes means that it is important to consider the power efficiency, or link budget and size of the receiving aperture. This is dominated by the solid angle subtended by the boresight null of each OAM mode in the multiplex. For simplicity, rotational symmetry can be assumed, and the null size described as the divergence angle, which is the angle between the boresight and the direction to maximum gain. For a given transmitter aperture, the larger the OAM mode number, the larger the size of the null. The relationship between the mode number and the null size has been studied for optical systems [11], with the relationship appearing to hold

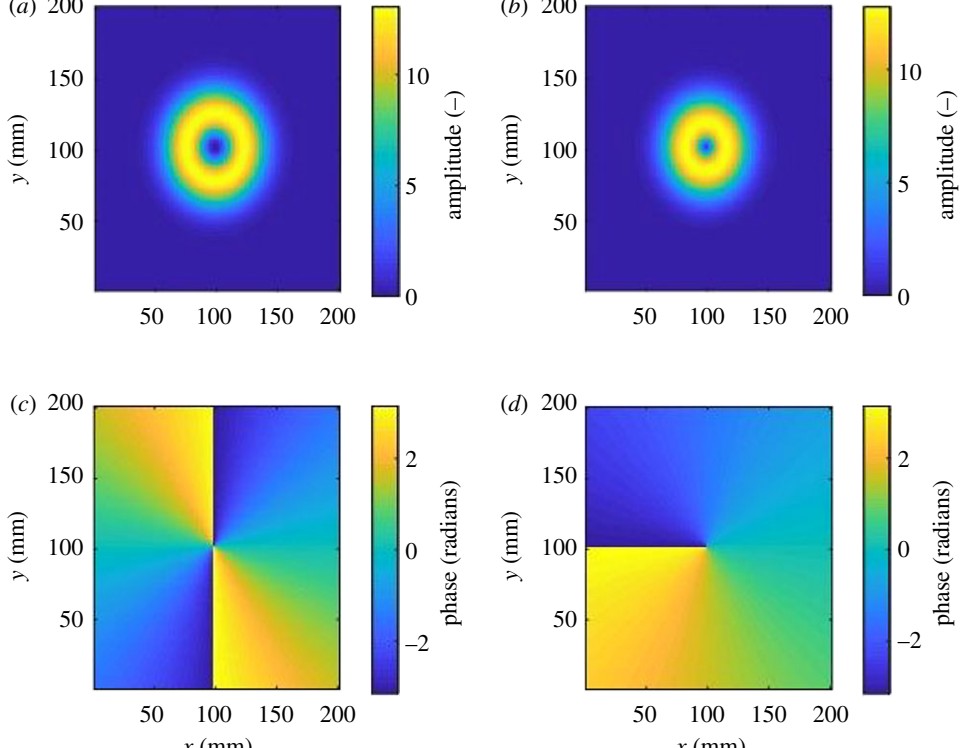

**Figure 1.** Profiles of LG modes amplitude ($a,b$) and phase ($c,d$), $l = 2$ ($a,c$) and $l = -1$ ($b,d$).

even with the smaller aperture sizes typical of radio transmitters [12]. The relationship between the divergence angles $\alpha_m$, $\alpha_n$ of two modes with mode numbers $l_m$, $l_n$ is

$$\frac{\alpha_n}{\alpha_m} = \left(\frac{l_n}{l_m}\right)^r, \tag{1.1}$$

where $r$ depends on the standard deviation of the radial intensity variation of the beam (a characteristic of the transmitter design) and has the limiting values of $r = 0.5$ (best case) and $r = 1.0$ (worst case). The receiving antenna must then be large enough to intercept at least some, preferably most, of the power transmitted in each OAM mode. Hence, the maximum link distance is limited by the largest mode used in the multiplex because it has the largest divergence. For the sake of illustration, to intercept approximately half the power in the highest numbered OAM mode in the multiplex, $l_{max}$, the receiver antenna diameter $D_{rx}$ must extend out to the radius at which the mode reaches it maximum field strength. This depends on the link distance $d$ and is given by

$$D_{rx} = 2d \tan \alpha_{max}. \tag{1.2}$$

Another way of viewing the problem of a diverging mode is to consider the additional apparent path loss as a function of link distance, for a fixed transmitter and receiver aperture. In such a case, with a diverging mode, OAM links can appear to have path loss proportional to $d^4$ [13]. This does not imply that the links are a near-field effect, but rather that received power falls more rapidly than with a typical boresight beam, because for OAM modes, as the distance increases, the receiver intercepts less of the power in the mode. Therefore, unless one uses receivers with rather large dimensions, it is imperative to reduce the divergence of the transmitted modes such as by using lenses.

To date, three principal means of generating and receiving OAM modes at radio frequencies have been proposed:

1. beamforming using a circular array [14],
2. spiral phase plates (SPPs) [15–17], and
3. ring resonator [18,19].

The focus of this paper is on the evaluation of SPPs. These consist of a dielectric structure that transforms an illuminating Gaussian beam into the spiral phase LG beam. They have a spiral optical thickness

profile that determines the spatial phase profile radiated from the device. The spiral may be continuous or approximated by using steps. There is a large step between the start and end of the spiral (figure 5a), where the height of the step, $h_t$, is determined by

$$h_t = \frac{l\lambda}{(n_1 - n_2)},$$ (1.3)

where $l$ is the mode number, $\lambda$ is the wavelength, $n_1$ is the refractive index of the SPP material and $n_2$ is the refractive index of the surrounding media.

In comparison to the work in [15], we evaluate SPPs that have been three-dimensionally (3D) printed using high-permittivity materials and report on the collimation effect produced by combining them with a 3D printed dielectric lens. The remainder of this paper is organized as follows. Section 2 describes a means of benchmarking OAM modes, §3 describes the design and realization of the SPPs, lens and horn. Section 4 describes the measurements of the SPPs, including the use of a dielectric lens for collimating the beams and the discussion the results. Conclusions are drawn in §5.

## 2. Benchmarking OAM modes

To evaluate the performance of our 3D printed horn antenna and SPPs, we compare them to an estimate of the link performance from a close-to-ideal system with a similar aperture size. An arbitrary array of 30 electric current sources arranged in a uniform concentric ring array of three concentric rings of 4, 10 and 16 elements (innermost to outermost) and spaced $\lambda/2$ apart, with a maximum radius of $1.25\lambda$ is used to generate OAM modes. This size and type of array was chosen to approximate the aperture of the 3D printed horn and SPP. The directivity and mode purity of these close-to-ideal beams can then be computed and used for comparison with data measured from experimental devices. To model the lens correctly, a larger arbitrary array of 270 elements in 9 concentric rings spaced $\lambda/2$ apart, with a maximum radius of $2.25\lambda$ was required.

The arbitrary sources used to model the horn/SPP aperture and lens aperture are normalized electric current sources with their far-field pattern truncated to the forward 'visible' half space. The OAM mode $l$ is then excited across the arbitrary array by phase weighting $w(n)$ each element $n$ according to its $\varphi$ position $\phi_n$ about the array normal, which is aligned with $(\theta, \phi) = (0,0)$

$$w(n) = e^{il\phi_n}.$$ (2.1)

The benefit of this approach is that it allows rapid assessment to be made on the optimum directivity and mode purity which can be expected from an aperture of a certain size and frequency. However, it does not represent a complete electromagnetic model of an array, with no mechanism for modelling inter-element scattering or environmental scattering. This level of detail falls outside the scope of this paper.

In order to characterize the OAM mode purity, the mode divergence angle ($\alpha_l$) and mode directivity, the far-field antenna pattern, $F(\theta, \phi)$, is decomposed via an angular Fourier transform into a set of coefficients $c_m$ based upon the fundamental helical modes $e^{il\phi}$, [20]

$$c_l(\theta) = \frac{1}{2\pi} \sum_{n=0}^{N-1} F(\theta, \phi_l) \cdot e^{-il\phi_n}.$$ (2.2)

The relative power of each mode $P(l)$ can then be calculated from the coefficients by integrating their squared modulus along the $\theta$ axis, for which $S$ is the total far-field power contained within all modes for that far-field pattern. If the sum contains all modes for both the co-polar and cross-polar components of the far-field pattern, then both the co-polar and cross-polar mode spectra can be found relative to the total power in all modes

$$\chi = \sum_l \int_\theta d\theta |c_l(\theta)|^2.$$ (2.3)

The maxima of the desired mode coefficients $(c_l(\theta)_{max})$ will be the divergence angle ($\alpha_l$) of mode $l$, and this can then be used to calculate the appropriate link distance $d$ for a pair of apertures of diameter $D_{rx}$

$$P(l) = \frac{1}{\chi} \int d\theta |c_l(\theta)|^2.$$ (2.4)

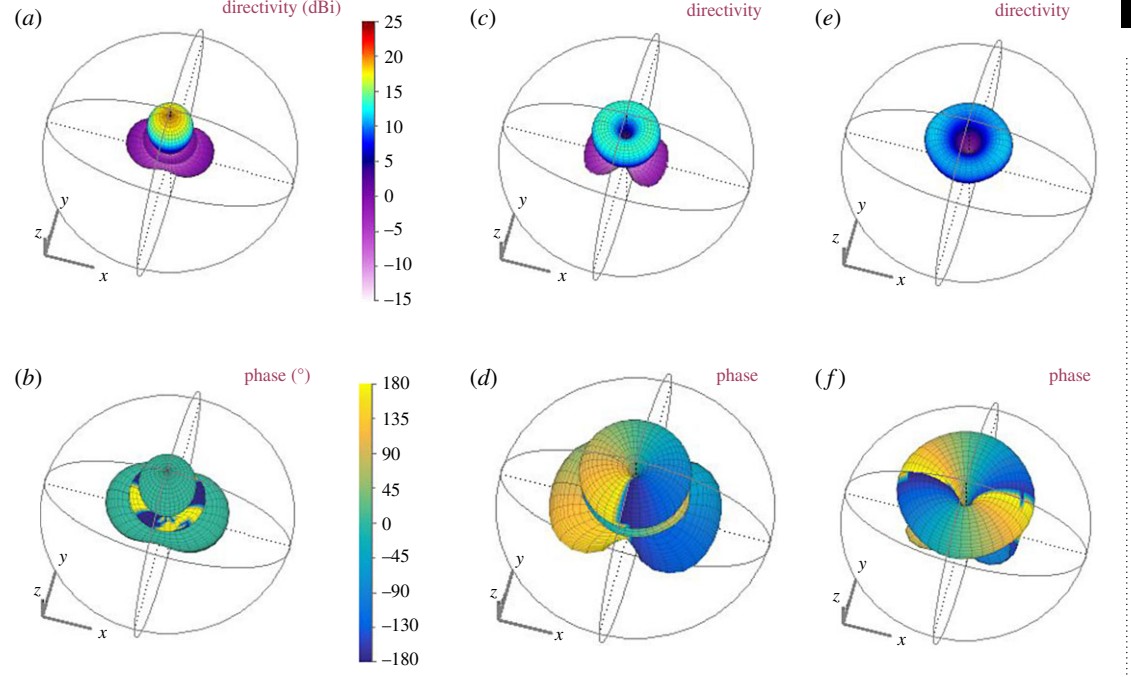

**Figure 2.** Simulated modes 0, 1 and 2. For the phase patterns, the phase has been superimposed onto the directivity pattern. The conical nature of the beams is evident for modes 1 and 2. (*a*) Mode 0 directivity pattern, (*b*) mode 0 phase pattern, (*c*) mode 1 directivity pattern, (*d*) mode 1 phase pattern, (*e*) mode 2 directivity pattern and (*f*) mode 2 phase pattern.

Figure 2 shows the resulting 3D radiation patterns for modes 0, 1 and 2. The directivity and phase patterns are shown for each, where the phase has been superimposed onto the directivity pattern. For the case of mode 0, a Gaussian main beam is shown along with sidelobes, and the corresponding phase pattern exhibits a near-constant phase across the main lobe. The phase patterns corresponding to modes 1 and 2 exhibit a vortex in the middle. In each case, the characteristic phase variation can be observed around the central vortex where the direction and rate of phase change is determined by the mode. Figure 3*a* shows directivity for each mode across a 'slice' of the 3D pattern and for angles of $\varphi = 0$–$90°$, i.e. the pattern is assumed to be symmetric across the *y*-plane. This shows the widening of the vortex as the mode order increases. The mode spectrum plots in figure 3*b* have been determined by plotting $\chi$ and used to determine the level of crosstalk between mode 0 to modes $\pm 2$, which gives $-27$ dB; from mode 1 to modes $-1$ and $+3$ of $-24$ dB; from mode 2 to modes 0 and $+4$ of $-24$ dB, and from mode 2 to mode $-2$ of $-20$ dB. The crosstalk matrix is shown in tables 1 and 2 for incident modes $l = 0, 1, 2$ and receive modes for $l = -2, -1, 0, 1, 2$. Table 2 shows the maximum achievable spectral efficiency (SE) for each mode in bits $s^{-1}$ $Hz^{-1}$, where up to 10 bits $s^{-1}$ $Hz^{-1}$ can be achieved. The total spectral efficiency for the multiplex is the summation of all SEs in the multiplex [21], i.e. 111 bits $s^{-1}$ $Hz^{-1}$, which gives a maximum achievable channel capacity of 111 Gb $s^{-1}$ assuming a 1 GHz bandwidth. SE has been computed from the crosstalk levels in table 1 and using the Hartley–Shannon's channel capacity formula as follows:

$$\mathrm{SE} = \log_2\left(1 + \frac{S}{I}\right),\tag{2.5}$$

where $S$ is the signal power, $I$ is the interference power and thermal noise is assumed negligible compared to $I$. The above expression holds for band-limited signals with average white Gaussian noise under a mean power constraint and a memoryless channel. It is a topic for future investigation to determine the crosstalk statistics and refine the SE calculation. According to the above analysis, spectrally efficient modulation schemes[1] of up to 256QAM may be supported.

---

[1]A spectrally efficiency modulation scheme is the one that increases the number of bits sent per unit of time without increasing the signal bandwidth. Examples are: QPSK, 64QAM, etc. The alternative is power-efficient modulation schemes that do not require additional transmit power for transmitting more data per unit time but instead require more bandwidth. Examples are FSK, 4FSK, 16FSK, etc.

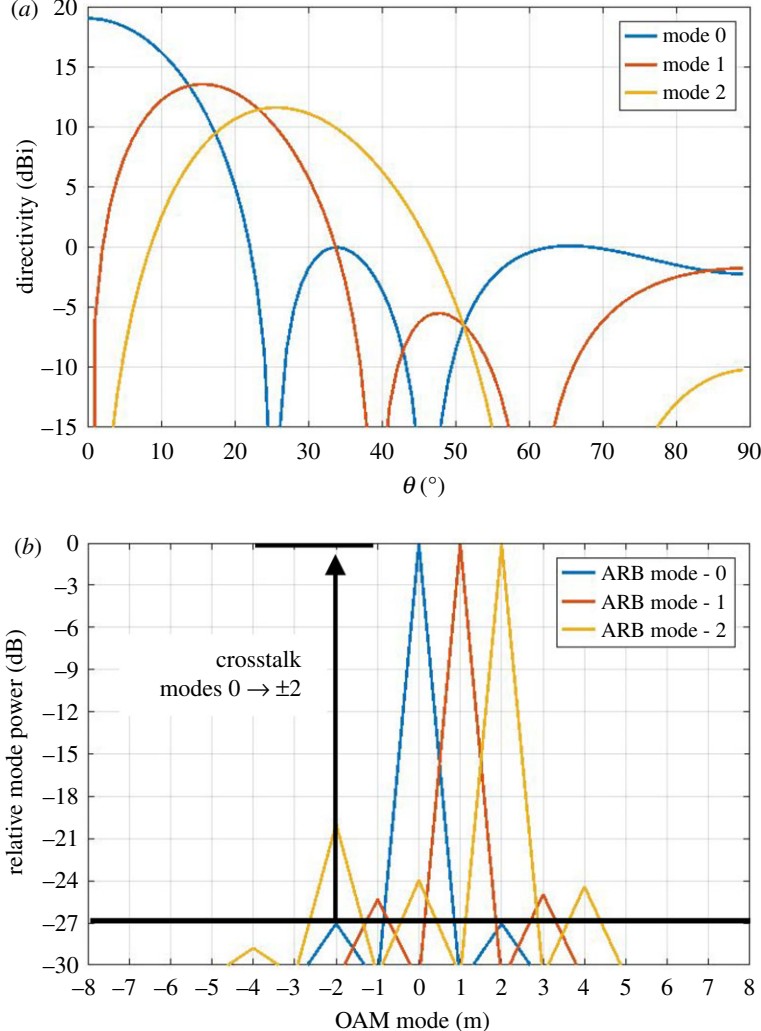

**Figure 3.** (*a*) Directivity patterns for modes 0, 1 and 2 for a 'cut' across the centre of the 3D pattern. (*b*) Resulting mode spectra from the simulated radiation patterns.

**Table 1.** Crosstalk matrix for arbitrary array model, levels shown in dB.

| incident mode | received mode | | | | |
|---|---|---|---|---|---|
| | −2 | −1 | 0 | 1 | 2 |
| 0 | −27 | −30 | — | −30 | −27 |
| 1 | −30 | −26 | −30 | — | −30 |
| 2 | −20 | −30 | −24 | −30 | — |

**Table 2.** Modal spectral efficiency in bits s$^{-1}$ Hz$^{-1}$.

| incident mode | received mode | | | | |
|---|---|---|---|---|---|
| | −2 | −1 | 0 | 1 | 2 |
| 0 | 9.0 | 10.0 | | 10.0 | 9.0 |
| 1 | 10.0 | 8.6 | 10.0 | | 10.0 |
| 2 | 6.7 | 10.0 | 8.0 | 10.0 | |

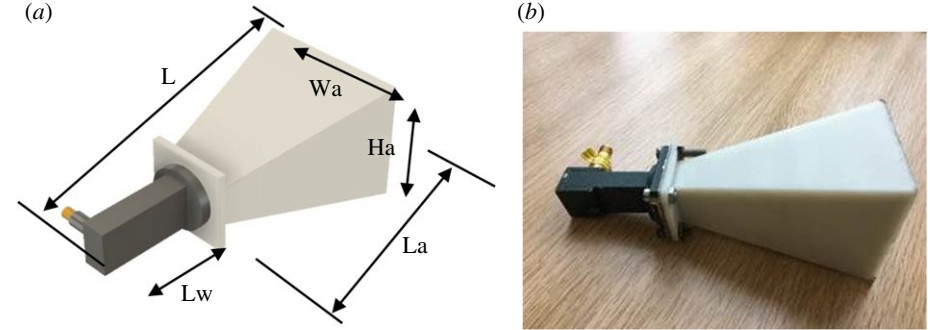

**Figure 4.** (*a*) Schematic drawing of the horn antenna. (*b*) Photo of 3D printed horn antenna.

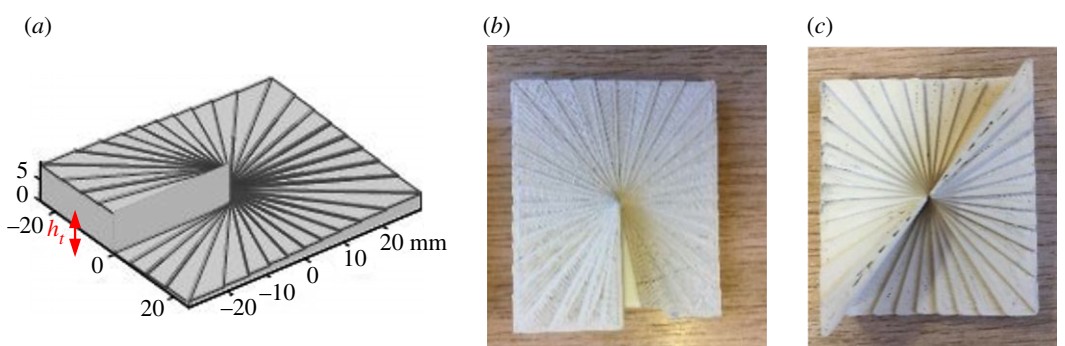

**Figure 5.** SPP design and realization. (*a*) SPP mode 1 dimensions, (*b*) photo of printed SPP mode 1 and (*c*) photo of printed SPP mode 2.

**Table 3.** Waveguide and horn outside dimensions.

| parameter | value |
|---|---|
| overall length (L) | 123 mm |
| aperture length (La) | 87.1 mm |
| aperture width (Wa) | 43.18 mm |
| aperture height (Ha) | 54.07 mm |
| waveguide length (Lw) | 47 mm |

# 3. Horn, spiral phase plate and lens design and realization

## 3.1. Horn design and realization

A pyramidal horn antenna was designed with a centre frequency of 15 GHz and gain of 18.3 dBi, and fabricated using a material extrusion dual-head desktop 3D printer (Makerbot Replicator2X) and a commercial acrylonitrile butadiene styrene filament (ABS, $\varepsilon_r = 2.57$, tan$\delta = 0.0047$). The wall thickness of the printed horn antenna was 2 mm. Conducting copper paint (Caswell Plating) was used for metallization. The horn antenna was connected to a Ku band coaxial–waveguide transition. Figure 4*a* shows the printed horn, while figure 4*b* shows the horn connected to the waveguide transition, and the key dimensions are listed in table 3.

## 3.2. Spiral phase plate design and realization

The SPPs for modes 1 and 2 were designed for an operating frequency of 15 GHz and comprised a spiral stepped 'staircase' as shown in figure 5. The overall rectangular shape of the SPPs ensured the exit aperture

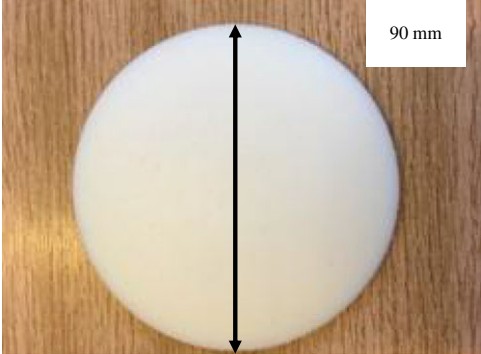

90 mm

**Figure 6.** Photo of the 90 mm diameter, 10 mm thick (maximum) 3D printed dielectric lens.

of the radiating horn antenna was covered. A quarter wavelength anti-reflection plate was printed in ABS and glued to the back of the SPP. The SPPs themselves were printed using a bespoke, in-house high-permittivity composite filament composed of barium titanate ($BaTiO_3$) nanoparticles embedded within an ABS matrix with $\varepsilon_r = 11 + j3.03 \times 10^{-2}$. Details of the material manufacturing and characterization can be found in [22]. The total height ($h_t$) of the SPP for mode 1 was 1.1 mm. Compared with conventional polymer-only dielectric printed SPPs ($\varepsilon_r = 2.6$), this represented a height reduction of approximately 75% for the same mode at the same operating frequency. Schematics of the SPP design and the printed SPPs are shown in figure 5.

The SPPs were printed using a fused deposition method. The continuous composite filament was fed into a tube where a pinch roll mechanism pushed the filament through a heating chamber. The composite filament was partially melted and deposited onto the build platform to form a layer-by-layer structure.

While the increasing thickness in the clockwise direction makes it clear that these SPPs are designed to induce positive OAM modes, it should be noted that the anechoic chamber used to measure the combined horn, SPP and lens far-field patterns is set up to use a reference horn as the transmitter and the test horn/SPP/lens to receive the test signals. The consequence is that the handedness (the circular progression of the spiral) of the SPPs is reversed compared to when the test horn/SPP/lens is used for launching signals making the spiral effect in the other direction.

## 3.3. Dielectric lens design and realization

An abi-convex dielectric lens was used to collimate the OAM radio beam and had a diameter of 90 mm, maximum thickness of 10 mm and was designed for a focal length of 110 mm. The lens was printed using a Stereo Lithography (SLA) 3D printer (Stratsys J750 Multijet) using a photo-curable resin ($\varepsilon_r = 3$) called 'VeroWhitePlus'. A photo of the printed lens is shown in figure 6.

# 4. Measurements, results and discussion

## 4.1. Measurement configuration and procedure

Three-dimensional complex radiation patterns have been measured for the following combinations:

— horn,
— horn/SPP,
— horn/lens, and
— horn/SPP/lens.

This was repeated for OAM modes 1 and 2. The measurements took place in an anechoic chamber and used an Anritsu 37397C Vector network analyser set to 15 GHz. This reference signal was first amplified and then emitted from a Flann DP240 horn antenna positioned 5 m from the unit under test (UUT). The UUT was mounted on a two-axis rotator and stepped around in 10° intervals over azimuth and elevation, which was repeated for both vertical and horizontal polarizations and for co- and cross-polar measurements. The measurement data were logged on a computer for subsequent analysis. A photo

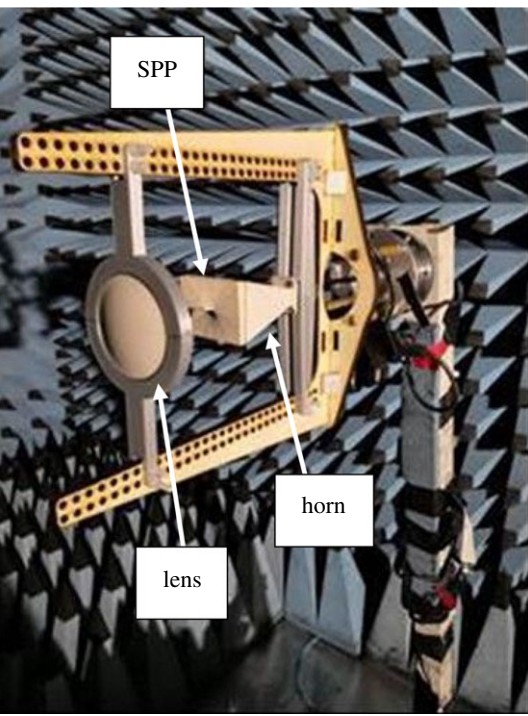

**Figure 7.** Horn, SPP and lens measurement configurations.

of the complete horn/SPP/lens combination inside the anechoic chamber is shown in figure 7. The lens was positioned using a rigid plastic frame with the following distances from the horn:

— 90 mm when there was no SPP,
— 54 mm for mode 1, and
— 34 mm for mode 2.

These distances were selected experimentally to maximize mode purity. This is explained in the following section and shows that there is a trade-off between maximizing mode purity and beam collimation.

## 4.2. Results

The complex 3D radiation patterns for each of the horn/SPP/lens combinations are shown in the electronic supplementary material. These have been used for further analysis of directivity and OAM mode purity, which enables the level of crosstalk between modes to be determined.

Figure 8*a* depicts a 2D azimuth 'cut' through the centre of the 3D directivity patterns for each horn/SPP/ mode combination. Due to symmetry, only 0°–90° is shown. With reference to 'a' in figure 8, an increase in directivity of 6 dBi is shown for when the lens is placed in front of the horn. With a mode 1 SPP in place, the directivity is shown to decrease by 4.2 dB (b). By placing the lens in front, the directivity is increased by 1.9 dB and the sidelobes are shown to slightly decrease for both modes 1 and 2 (c and d), indicating a beneficial effect of the lens. However, it is notable from the figure that when an SPP is in place, there is an increase in the overall system loss, which would be the case at each end of a point-to-point communications link. For comparison, the loss due to the SPP in [15] is reported to be 5.5 dB per SPP.

The mode spectrum for each of the horn/SPP/lens combinations has been determined from the 3D phase patterns, as described in §1. The resulting spectra are shown in figure 8*b*. By way of example, assume the two modes of interest are $l = \pm1$. Thus, the signal assigned to mode +1 will experience −11 dB of crosstalk from the signal assigned to mode −1 (a). This compares to a crosstalk of −24 dB shown in the benchmark in figure 3. When the lens is included, this is reduced to −8 dB. This compares with −13 dB achieved in [15], where much larger continuous spiral SPPs were used. The resulting crosstalk matrix for incident modes of 0, 1 and 2; and receive modes of −2, −1, 0, 1 and 2 are shown in tables 4 and 6 for without and with a lens (except mode 0 as this was not measured),

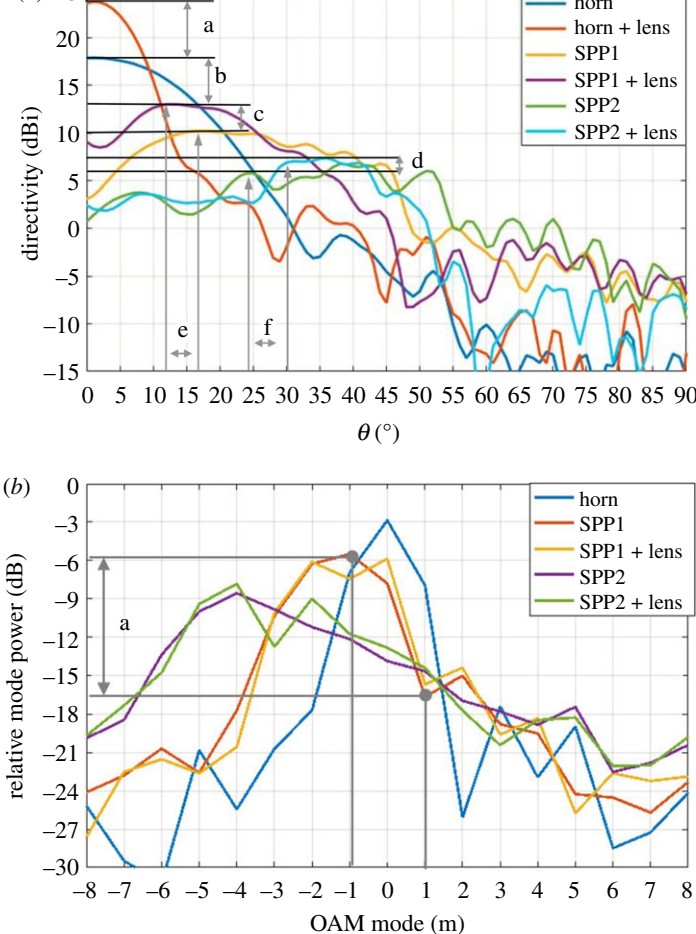

**Figure 8.** (*a*) Two-dimensional 'cut' showing measured total directivity patterns of several horn/SPP/lens configurations. Labels: a is the increase in directivity due to lens; b is the loss of directivity due to SPP1; c is the increase in directivity due to SPP1 and lens; d is the increase in directivity due to SPP2 and lens; e is the change in the angle of maximum directivity of mode 1 due to inserting lens; f is the change in angle of maximum directivity of mode 1 due to inserting lens. (*b*) Mode spectrum for measured 3D printed horn, SPPs and optimized lens separation distance. a is the difference in modal powers between modes −1 and +1.

respectively. This difference in modal crosstalk is likely to be due to the distortion in the far-field phase gradient caused by the inability of the lens to capture a large enough proportion of the power in the 'doughnut' shaped maximum directivity region, where the mode is also purest. This is explained by considering the geometry of the lens radius, focal length and the angle of maximum directivity of OAM modes 1 and 2. This reveals that mode 2 cannot be optimally supported with this lens and thus, a larger diameter lens or suboptimal focal length is selected that minimizes crosstalk. The latter is the case implemented in this work.

To examine mode purity in more detail, figure 9 is included that shows the mode 0 and mode −1 coefficients for the mode 1 SPP with and without the lens as a function of angle. Without the lens, the mode −1 region can be clearly observed in the region $18° \leq \theta \leq 25°$ (blue curve), where a strong mode 0 component is observed (orange curve). When the lens is included, the mode −1 region shifts to $15° \leq \theta \leq 23°$, reducing the divergence angle, but the mode 0 component is increased in power to near parity with the desired component over the angular region of $0° \leq \theta \leq 15°$ (purple curve). The corresponding effect of this on the OAM mode spectrum can be clearly seen in figure 9 as the mode spectrum for mode 1 is quite broad. This effect is even more pronounced for the mode 2 SPP, due to the increased divergence angle compared to the mode 1 SPP. In this case, the symmetric mode crosstalk is −5.7 dB without the lens, and −11.4 dB with the lens. The maximum directivity is increased by 0.85 dB by the addition of the lens. In both cases, the lens is increasing the maximum directivity while also decreasing the mode purity, as the adjacent mode power increased by 2 dB (1–0 coupling) and 4 dB (1–2 coupling) with the addition of the lens (tables 4–7). In effect, the lens is collimating the region of the far field for which the mode 0 coefficients dominate, rather than the

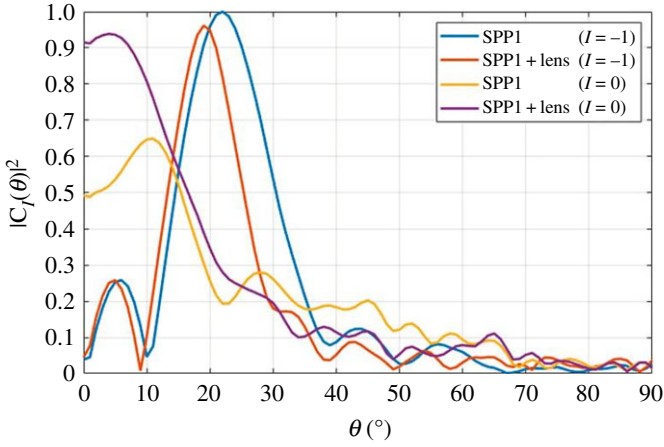

**Figure 9.** Dielectric lens and its effects on OAM mode purity and divergence angle.

**Table 4.** Crosstalk matrix without lens, levels shown in dB.

| | received mode | | | | |
|---|---|---|---|---|---|
| incident mode | −2 | −1 | 0 | 1 | 2 |
| 0 | −25 | −8 | — | −7 | −18 |
| 1 | −15 | −17 | −8 | — | −6 |
| 2 | −17 | −15 | −14 | −12 | — |

**Table 5.** Spectral efficiency matrix without lens, levels shown in bits s$^{-1}$ Hz$^{-1}$.

| | received mode | | | | |
|---|---|---|---|---|---|
| incident mode | −2 | −1 | 0 | 1 | 2 |
| 0 | 8.3 | 2.9 | — | 2.6 | 6.0 |
| 1 | 5.0 | 5.7 | 2.9 | — | 2.3 |
| 2 | 5.7 | 5.0 | 4.7 | 4.1 | — |

**Table 6.** Crosstalk matrix with lens, levels shown in dB.

| | received mode | | | | |
|---|---|---|---|---|---|
| incident mode | −2 | −1 | 0 | 1 | 2 |
| 1 | −15 | −16 | −6 | — | −2 |
| 2 | −18 | −12 | −13 | −14 | — |

**Table 7.** Spectral efficiency matrix with lens, levels shown in bits s$^{-1}$ Hz$^{-1}$.

| | received mode | | | | |
|---|---|---|---|---|---|
| incident mode | −2 | −1 | 0 | 1 | 2 |
| 1 | 5.0 | 5.4 | 2.3 | — | 1.4 |
| 2 | 6.0 | 4.0 | 4.3 | 4.7 | — |

annular region for which the desired mode coefficients dominate. To maintain appropriate isolation between each mode, an improved method of collimation is required.

Note that when positioned to maximize directivity for the mode 1 SPP, the lens was 54 mm from the horn aperture and 44.7 mm from the top of the SPP; while for the mode 2 SPP, the lens was 34 mm from the horn aperture and 15.4 mm from the top of the SPP. These are not the optimum distances for maximum directivity from the lens, as shown by the poor increase in maximum directivity. However, these positions were chosen to maximize the proportion of the OAM mode captured by the lens and subsequently collimated. When the lens is positioned for maximum directivity, without an SPP, the maximum directivity is 24 dBi, an increase of 5.6 dB, as shown in figure 9. The arbitrary array model for the lens predicted an increase of 4.5 dB and a maximum directivity of 23.5 dBi, showing close agreement. It is this agreement between the arbitrary array and measured patterns for the horn alone that gives confidence to the arbitrary array predictions as to the upper limits on mode isolation and directivity for the horn/SPP/lens combination. In terms of total directivity, the horn aperture without the SPPs is close to the ideal for its size.

The crosstalk levels in tables 4 and 6 have been used to determine the spectral efficiency per mode by using the same method described in §2. The resulting spectral efficiencies are given in table 5 for without a lens and table 7 with a lens. These give a spectral efficiency for the total multiplex of 55 bits s$^{-1}$ Hz$^{-1}$ without a lens and 33 bits s$^{-1}$ Hz$^{-1}$ with the lens, giving a maximum achievable data rate of 55 Gb s$^{-1}$ and 33 Gb s$^{-1}$, respectively, assuming a 1 GHz bandwidth. According to the tables, modulation schemes of MFSK through to 256QAM may be supported depending on the crosstalk for each mode. We can see that the capacity is reduced when the lens in used, although directivity is enhanced and indicates considerable room for performance improvement.

## 4.3. Discussion

From the results, we can see that loss, beam divergence and collimation are three issues that need to be tackled for the application of SPPs to be viable for OAM multiplexing a radio link. The insertion loss of the SPPs occurs at both ends of the link, as is the case for the lens and mode multiplexer. We suggest that the insertion loss of the SPPs may be reduced by:

— operating at a higher frequency because the SPPs will be thinner;
— using a lower loss dielectric while retaining a high dielectric constant, such as a ceramic material;
— including an impedance matching surface on the front and back of the SPP matched individually to each segment of the spiral phase pattern.

Minimizing crosstalk is vital for supporting high data rate communications due to the lower level of signal-to-noise plus interference ratio demanded by higher order QAM modulation schemes such as those used by Wi-Fi, LTE and forthcoming 5G devices. In such case, crosstalk levels of less than −30 dB are required to achieve the highest modulation/coding levels. This is proving a challenge when using SPPs and, at the time of writing, the authors are not aware of any works that approach this level without using interference cancellation signal processing techniques. With careful design and realization, improvements are possible, but the required step-change in performance may be difficult to achieve. We suggest that optimum crosstalk can be achieved by careful selection of the distance from the SPP to the lens for each mode. This may be implemented as shown in figure 10 and scaled when more than two modes are used. The consequence of this is that the incident energy profile on the lens can be adjusted and compensate for non-ideal radiation patterns. It also enables novel modes to be produced, i.e. in an ideal set-up, the magnitude and phase profiles of two modes are distinct, yet by using this technique, the magnitude profile of two modes can be adjusted to overlap, while the two modes would still be distinguishable by their phase profiles. An alumina plate is shown for beam splitting/combining. This can be considered as a ½ mirror and therefore is only 50% efficient, which is also the case for the multiplexer used in [15]. As for the SPPs, this may be improved by including an impedance matching surface on both sides, and the same can also be incorporated into the lens. This is expected to dramatically reduce the loss experience in this choice of OAM implementation.

Beam collimation is also vital as this limits beam divergence and hence determines the achievable link distance. The lens aperture should be enough to capture all the radiated energy from all modes. It should also be constructed using low-loss dielectric material and include matching surfaces. Our results also highlight a trade-off between collimation, mode purity (which determines crosstalk)

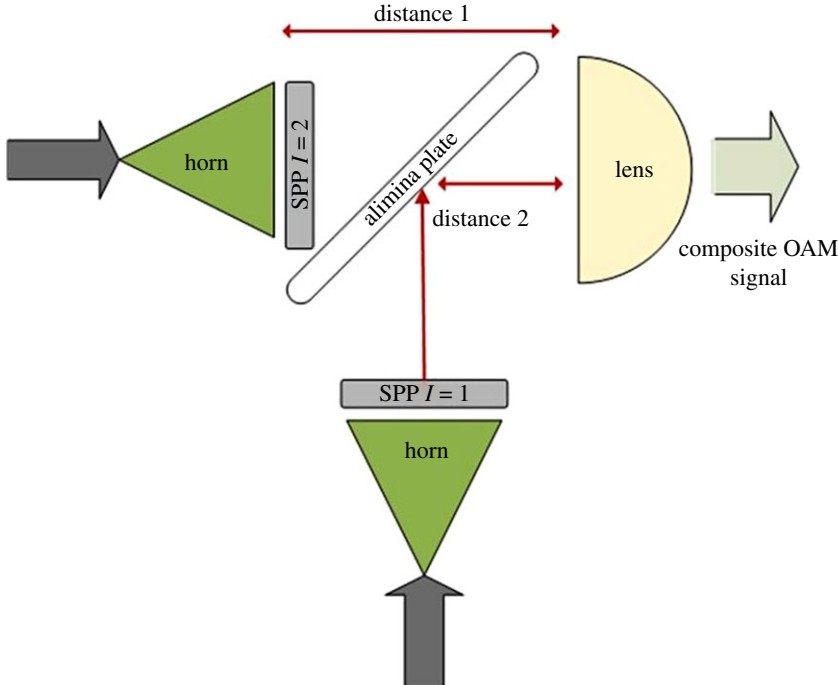

**Figure 10.** Architecture of an SPP mode multiplexer with alumina plate ½-mirror and optimal distances for modes 1 and 2.

and spectral efficiency, thus highlighting the need for a lens optimized for maximizing both mode purity and collimation.

# 5. Conclusion

We have presented an experimental evaluation of using 3D printed SPPs for producing OAM radio signals. The results with the lens yielded a crosstalk of −8 dB between modes 1 and −1, and −11.4 dB between modes 2 and −2. An additional loss of 4.2 dB has been shown when the SPPs are inserted into the communication link, which is undesirable for obtaining reliable LTE-based communications. Insertion loss, beam divergence and mode purity remain challenges when using SPPs for OAM mode multiplexing, and, through careful design, there is much performance potential to be attained towards reaching the theoretical performance limit. We have suggested practical ways to reduce loss, increase mode purity and reduce beam divergence. For example, we have suggested a mode multiplexer architecture that is expected to further reduce the crosstalk for each mode. We have also suggested using lower loss materials and to seek ways to reduce material interface reflections. We have shown that 3D printing and additive manufacturing is a viable and attractive means of rapid prototyping of microwave horns, SPPs and lenses. The benefit to continued investment of effort in improving an SPP-based approach is the cost compared to using an electronic beam-forming approach.

Data accessibility. Measured complex radiation pattern data used to produce the results in §4 are available from doi:10. 5523/bris.3juehd5wwl6491zzz99038tll9 and http://data.bris.ac.uk/data/dataset/3juehd5wwl6491zzz99038tll9 [23].

Authors' contributions. B.A. drafted the manuscript, guided the work and performed some of the analysis. T.P. drafted §§2 and 4.2 and performed the measurements and analysis. Y.W. contributed the spiral phase plates, horns and drafted §§3.1 and 3.2. T.D. drafted much of the introduction and background theory. D.I. contributed to the lens. C.G. contributed to the measurements and analysis and critically reviewed the manuscript. C.J.S. has made substantial contributions to the concept and has critically reviewed the manuscript. G.H. facilitated the measurements and critically reviewed the manuscript. M.A.B. facilitated the measurements and critically reviewed the manuscript. P.S.G. conceived the manufacturing approach and the development of the bespoke materials, and edited the manuscript. All authors gave final approval for publication.

Competing interests. We declare we have no competing interests.

Funding. B.A. is grateful for the support of the Royal Society Industrial Fellowship scheme under award no. IF160001. P.S.G. and Y.W. acknowledge funding by the UK Engineering and Physical Science Research Council (grant no. EP/ P00578/1).

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
