## [Reviewer comments · Royal Society Open Science]

Review History

RSOS-191419.R0 (Original submission)

Review form: Reviewer 1

Is the manuscript scientifically sound in its present form?

Yes

Are the interpretations and conclusions justified by the results?

Yes

Is the language acceptable?

Yes

Do you have any ethical concerns with this paper?

No

Have you any concerns about statistical analyses in this paper?

No

Recommendation?

Accept with minor revision (please list in comments)

Comments to the Author(s)

Some minor comments:

The authors presented the PPS and lens for OAM generation, and implementation and measurement results are provided.

1. The authors are suggested to correct some grammatically issues in the paper.

'We suggest a mode multiplexer architecture that is expected to further reduce the crosstalk for each mode is employed.' Which appeared twice in the paper.

In summary, 'seek ways to reduce material' should be 'seeking ways ...';

'topological charge' should be 'topological change'

'This compares with 13 dB achieved', should it be -13 dB?

On page 10, 'with this lens and this a larger diameter lens or sub ...', please rewrite.

2. The authors use 'S' to represent two different parameters in the paper, please change one of them.

Review form: Reviewer 2 (Anas Mohsin)**Is the manuscript scientifically sound in its present form?**

Yes

Are the interpretations and conclusions justified by the results?

Yes

Is the language acceptable?

Yes

Do you have any ethical concerns with this paper?

No

Have you any concerns about statistical analyses in this paper?

No

Recommendation?

Accept with minor revision (please list in comments)

Comments to the Author(s)

Use of OAM in the millimetre wave (mmwave) regime is certainly a promising approach to deliver increased spectral efficiency, especially given that both OAM and mmwave require line-of-sight to achieve meaningful gain in system capacity. The paper is very well-written and easy to

follow. I therefore recommend this paper to publications. However, I have minor comments to the authors which can benefit those who are interested in this subject.

- 1- The authors discussed multiple modes of OAM, how do these modes differ? do they share the same wave number, dispersion and wave-impedance or are they fundamentally different?
- 2- It is not clear if MIMO precoding can play a role in mitigating the encountered cross-talk. Have the authors considered the use of the eigen-modes of OAM if there're any?
- 3- Given that OAMs are non-plane waves, is there a scope for use of reflective materials without comprising the orthogonality of OAMs in non-line of sight channels?
- 4- On the crosstalk levels, would the use of spatial filters (digital eg using the null-space) help in reducing crosstalk, this is related to comment no 2
- 5- Page 4, first line, it's grammatically more accurate to use "that" in "means it is important..."
- 6- In Page 3, lines 39-42, spaces are required between the number and the start of the lines
- 7- The authors are recommended to include a channel scattering matrix to demonstrate the individual coupling coefficients for a sample frequency with and without lens to improve readability
- 8- Equation (4) seems fundamental to the orthogonality of OAMs, although the authors discuss angular fourier transform, it's not entirely clear equation (5) dictates equation (4), would be useful to clarify this. As a reader, I expect the optimal solution for equation (4) is fourier-based.
- 9- It's clear that the lens improved the channel gain of the diagonal channel coefficients however the impact on the off-diagonal is unclear. The authors may want to rephrase, addressing (7) will resolve this point.

Overall, the paper is definitely worth publishing and the subject is timely, relevant and stimulating.

Decision letter (RSOS-191419.R0)

28-Oct-2019

Dear Dr Allen

On behalf of the Editors, I am pleased to inform you that your Manuscript RSOS-191419 entitled "Experimental Evaluation of 3D Printed Spiral Phase Plates for enabling an Orbital Angular Momentum Multiplexed Radio System" has been accepted for publication in Royal Society Open Science subject to minor revision in accordance with the referee suggestions. Please find the referees' comments at the end of this email.

The reviewers and handling editors have recommended publication, but also suggest some minor revisions to your manuscript. Therefore, I invite you to respond to the comments and revise your manuscript.

- Ethics statement

- Data accessibility

<http://datadryad.org/submit?journalID=RSOS&manu=RSOS-191419>

- Competing interests

- Authors' contributions

- Acknowledgements

- Funding statement

Please ensure you have prepared your revision in accordance with the guidance at <https://royalsociety.org/journals/authors/author-guidelines/> -- please note that we cannot publish your manuscript without the end statements. We have included a screenshot example of

the end statements for reference. If you feel that a given heading is not relevant to your paper, please nevertheless include the heading and explicitly state that it is not relevant to your work.

Because the schedule for publication is very tight, it is a condition of publication that you submit the revised version of your manuscript before 06-Nov-2019. Please note that the revision deadline will expire at 00.00am on this date. If you do not think you will be able to meet this date please let me know immediately.

Please note that Royal Society Open Science charge article processing charges for all new submissions that are accepted for publication. Charges will also apply to papers transferred to Royal Society Open Science from other Royal Society Publishing journals, as well as papers

submitted as part of our collaboration with the Royal Society of Chemistry (<http://rsos.royalsocietypublishing.org/chemistry>).

Kind regards,
Anita Kristiansen
Editorial Coordinator
Royal Society Open Science
openscience@royalsociety.org

on behalf of Professor Weisi Guo (Associate Editor) and R. Kerry Rowe (Subject Editor)
openscience@royalsociety.org

Reviewer comments to Author:
Reviewer: 1

Comments to the Author(s)

Some minor comments:

The authors presented the PPS and lens for OAM generation, and implementation and measurement results are provided.

1. The authors are suggested to correct some grammatically issues in the paper.

'We suggest a mode multiplexer architecture that is expected to further reduce the crosstalk for each mode is employed.' Which appeared twice in the paper.

In summary, 'seek ways to reduce material' should be 'seeking ways ...';

'topological charge' should be 'topological change'

'This compares with 13 dB achieved', should it be -13 dB?

On page 10, 'with this lens and this a larger diameter lens or sub ...', please rewrite.

2. The authors use 'S' to represent two different parameters in the paper, please change one of them.

Reviewer: 2

Comments to the Author(s)

Use of OAM in the millimetre wave (mmwave) regime is certainly a promising approach to deliver increased spectral efficiency, especially given that both OAM and mmwave require line-of-sight to achieve meaningful gain in system capacity. The paper is very well-written and easy to

follow. I therefore recommend this paper to publications. However, I have minor comments to the authors which can benefit those who are interested in this subject.

- 1- The authors discussed multiple modes of OAM, how do these modes differ? do they share the same wave number, dispersion and wave-impedance or are they fundamentally different?
- 2- It is not clear if MIMO precoding can play a role in mitigating the encountered cross-talk. Have the authors considered the use of the eigen-modes of OAM if there're any?
- 3- Given that OAMs are non-plane waves, is there a scope for use of reflective materials without comprising the orthogonality of OAMs in non-line of sight channels?
- 4- On the crosstalk levels, would the use of spatial filters (digital eg using the null-space) help in reducing crosstalk, this is related to comment no 2
- 5- Page 4, first line, it's grammatically more accurate to use "that" in "means it is important..."
- 6- In Page 3, lines 39-42, spaces are required between the number and the start of the lines
- 7- The authors are recommended to include a channel scattering matrix to demonstrate the individual coupling coefficients for a sample frequency with and without lens to improve readability
- 8- Equation (4) seems fundamental to the orthogonality of OAMs, although the authors discuss angular fourier transform, it's not entirely clear equation (5) dictates equation (4), would be useful to clarify this. As a reader, I expect the optimal solution for equation (4) is fourier-based.
- 9- It's clear that the lens improved the channel gain of the diagonal channel coefficients however the impact on the off-diagonal is unclear. The authors may want to rephrase, addressing (7) will resolve this point.

Overall, the paper is definitely worth publishing and the subject is timely, relevant and stimulating.

Author's Response to Decision Letter for (RSOS-191419.R0)

See Appendix A.

Decision letter (RSOS-191419.R1)

14-Nov-2019

Dear Dr Allen,

It is a pleasure to accept your manuscript entitled "Experimental Evaluation of 3D Printed Spiral Phase Plates for enabling an Orbital Angular Momentum Multiplexed Radio System" in its

current form for publication in Royal Society Open Science. The comments of the reviewer(s) who reviewed your manuscript are included at the foot of this letter.

on behalf of Professor Weisi Guo (Associate Editor) and R. Kerry Rowe (Subject Editor)
openscience@royalsociety.org

Appendix A

Dear Royal Society Open Science editor and reviewers,

We would like to thank you for considering our paper and for the positive response. On behalf of all authors, I have worked through the comments and made changes to the paper where appropriate. I enclose the updated paper with changes highlighted as well as a 'clean' version. A response to each of the comments is given below.

I hope you find this response sufficient to warrant publication.

Kind regards,

Ben Allen and co-authors

Reviewer comments to Author:

Reviewer: 1

The authors presented the PPS and lens for OAM generation, and implementation and measurement results are provided.

1. The authors are suggested to correct some grammatically issues in the paper.

'We suggest a mode multiplexer architecture that is expected to further reduce the crosstalk for each mode is employed.' Which appeared twice in the paper. **Good spot. Change made.**

In summary, 'seek ways to reduce material' should be 'seeking ways ...'; **Good spot. Change made.**

'topological charge' should be 'topological change'

The term 'topological charge' is considered to be correct in the context of this paper. The term refers to an integer that describes the mode number. It stems from quantum physics, 'topological quantum charge'. In the case of OAM modes, it refers to the number of phase 'twists' the beam exhibits in a single wavelength, thus the angular phase velocity.

'This compares with 13 dB achieved', should it be -13 dB? **Good spot and correction made.**

On page 10, 'with this lens and this a larger diameter lens or sub ...', please rewrite. **Done.**

2. The authors use 'S' to represent two different parameters in the paper, please change one of them. **Good spot. The first usage of s has now been changed to χ .**

Thank you very much for taking time to review our manuscript. We hope you find our response sufficient to recommend publication.

Reviewer: 2

Use of OAM in the millimetre wave (mmwave) regime is certainly a promising approach to deliver increased spectral efficiency, especially given that both OAM and mmwave require line-of-sight to

achieve meaningful gain in system capacity. The paper is very well-written and easy to follow. I therefore recommend this paper to publications. **Thank you very much.**

However, I have minor comments to the authors which can benefit those who are interested in this subject.

1- The authors discussed multiple modes of OAM, how do these modes differ? do they share the same wave number, dispersion and wave-impedance or are they fundamentally different?

The modes differ in three ways, as follows.

i. As the mode order increases, the width of the vortex increases. In other words, the diameter of the 'doughnut' shape in the amplitude radiation pattern increases.

ii. As the mode order increases, the number of phase 'twists' around the centre of the phase radiation pattern increases.

iii. Modes can be negative or positive, which refers to the 'handedness' of the phase twist mentioned above.

OAM modes actually refer to a class of modes called Laguerre Gaussian modes (LG modes), which can be created from a combination of standard waveguide modes.

2- It is not clear if MIMO precoding can play a role in mitigating the encountered cross-talk. Have the authors considered the use of the eigen-modes of OAM if there're any?

MIMO processing does play a role. It can be used to reduce interference caused by modal crosstalk. However the focus of the work does in the paper is the antenna engineering to create OAM modes that are as clean as possible. MIMO processing can be used to mop up any residue interference.

One way of analysing OAM modes is by standard MIMO analysis, and thus by considering the eigenmodes that are generated. This method is suited for the case when OAM modes are generated / received by circular antenna arrays. This is not the approach taken in this work, but has been considered by others.

MIMO Waterfilling also has a place in optimally allocating resources across modes.

3- Given that OAMs are non-plane waves, is there a scope for use of reflective materials without comprising the orthogonality of OAMs in non-line of sight channels? **Yes there is. In fact, this is the approach taken by early OAM radio work. The experiment took place in Venice circa 2010 and used a parabolic dish that was split across the radius to create an OAM mode.**

4- On the crosstalk levels, would the use of spatial filters (digital eg using the null-space) help in reducing crosstalk, this is related to comment no 2

Please see response to your comment 2.

5- Page 4, first line, it's grammatically more accurate to use "that" in "means it is important..." **Good spot and correction made.**

6- In Page 3, lines 39-42, spaces are required between the number and the start of the lines. **Good spot and correction made.**

7- The authors are recommended to include a channel scattering matrix to demonstrate the individual coupling coefficients for a sample frequency with and without lens to improve readability

Please see tables 1-4 at the end of the manuscript. These are tables giving the crosstalk levels between modes for a range of circumstances. Whilst these aren't the channel scattering matrices, they do indicate similar information as that derived from the scattering matrix. We do not have these measurements and it would be tricky to obtain them. However it is a good suggestion that we should take on board for future work.

8- Equation (4) seems fundamental to the orthogonality of OAMs, although the authors discuss angular fourier transform, it's not entirely clear equation (5) dictates equation (4), would be useful to clarify this. As a reader, I expect the optimal solution for equation (4) is fourier-based.

Equation 4 describes the phase pattern for a particular OAM mode. On the other hand, equation 5 is an analytical tool that enables the mode spectrum for a particular phase pattern to be determined. Of course, an ideal set-up would have a single mode that results in a single line on the angular Fourier transform. This can also be spotted by examining (4). However in a realistic system (4) will not hold and the resultant is a summation of several modes (Fourier series). This unknown composite mixture of modes can be determined from (5). I hope this clarifies somewhat.

9- It's clear that the lens improved the channel gain of the diagonal channel coefficients however the impact on the off-diagonal is unclear. The authors may want to rephrase, addressing (7) will resolve this point.

Thank you for this comment. Please see our response to (7), which we hope addresses this point too. Please note that we haven't taken the scattering matrix approach in this work but it is a good suggestion to take forward for future work.

Overall, the paper is definitely worth publishing and the subject is timely, relevant and stimulating.

Thank you very much for this positive comment. We hope you find our response to your comments sufficient to recommend publication.